# Perioperative LiMAx Test Analysis: Impact of Portal Vein Embolisation, Chemotherapy and Major Liver Resection

**DOI:** 10.3390/biomedicines12020254

**Published:** 2024-01-23

**Authors:** Felix Rühlmann, Azadeh Azizian, Christian Moosmann, Markus Bernhardt, Jan Keck, Hannah Flebbe, Omar Al-Bourini, Ali Seif Amir Hosseini, Marian Grade, Thomas Lorf, Michael Ghadimi, Thorsten Perl, Jochen Gaedcke

**Affiliations:** 1Department of General, Visceral, and Paediatric Surgery, University Medical Centre, D-37075 Göttingen, Germany; felix.ruehlmann@med.uni-goettingen.de (F.R.); azadeh.azizian@med.uni-goettingen.de (A.A.); c.moosmann@stud.uni-goettingen.de (C.M.); markus.bernhardt@med.uni-goettingen.de (M.B.); jan.keck@med.uni-goettingen.de (J.K.); hannah.flebbe@med.uni-goettingen.de (H.F.); marian.grade@med.uni-goettingen.de (M.G.); thomas.lorf@med.uni-goettingen.de (T.L.); mghadim@uni-goettingen.de (M.G.); thorsten.perl@med.uni-goettingen.de (T.P.); 2Institute for Diagnostic and Interventional Radiology, University Medical Centre Göttingen, D-37075 Göttingen, Germanyali.seif@med.uni-goettingen.de (A.S.A.H.)

**Keywords:** LiMAx test, perioperative risk assessment, liver function test, postoperative liver failure, portal vein embolisation, major liver resection

## Abstract

Background: Postoperative liver failure (PLF) is a severe complication after major liver resection (MLR). To increase the safety of patients, clinical bedside tests are of great importance. However, limitations of their applicability and validity impair their value. Methods: Preoperative measurements of the liver maximum capacity (LiMAx) were performed in *n* = 40 patients, who underwent MLR (≥3 segments). Matched postoperative LiMAx was measured in *n* = 21 patients. Liver function was compared between pretreated patients (*n* = 11 with portal vein embolisation (PVE) and *n* = 19 patients with preoperative chemotherapy) and therapy naïve patients. The LiMAx values were compared with liver-specific blood parameters and volumetric analysis. Results: In total, *n* = 40 patients were enrolled in this study. The majority of patients (*n* = 33; 82.5%) had high preoperative LiMAx values (>315 µg/kg/h), while only seven patients (17.5%) had medium values (140–315 µg/kg/h), and none of the patients had low values (<140 µg/kg/h). A comparison of pretreated patients (with PVE and/or chemotherapy) and therapy naïve patients showed no significant difference in the preoperative LiMAx values (*p* > 0.05). The preoperative LiMAx values were significantly higher than the matched postoperative values on postoperative day 1 (*p* < 0.0001). A comparison between the expected and measured postoperative LiMAx showed a difference (≥10%) in 7 out of 13 patients (53.8%). After an initial postoperative decrease in the LiMAx, the patients without complications (*n* = 12) showed a continuous increase until 14 days after surgery. In the patients with postoperative complications, a decrease in the LiMAx was associated with a prolonged recovery. Conclusions: For patients undergoing MLR within the 0.5% rule, which is the clinical gold standard, the LiMAx values do not offer any additional information. Additionally, the LiMAx may have reflected liver function, but it did not deliver additional information regarding postoperative liver recovery. The clinical use of LiMAx might be relevant in selected patients beyond the 0.5% rule.

## 1. Introduction

The accurate monitoring of liver function has always been of great interest to clinicians treating patients with liver diseases. Specifically, in surgical departments, the monitoring of liver function has gained more and more relevance due to an increased number of MLRs, enabling a potentially curative therapeutic approach for patients with advanced oncological diseases. While chemotherapy and surgical techniques have improved in recent decades, insufficient postoperative residual liver function remains the main limitation of MLR [1]. The occurrence of postoperative liver failure (PLF) is associated with high mortality and is crucial to avoid [2,3,4,5]. While hepatectomies of up to 80% of the liver volume are feasible in patients with healthy liver parenchyma, PLF can unexpectedly occur in predamaged liver tissue, such as a liver affected by steatosis, fibrosis, cirrhosis, cholestasis or chemotherapy-exposed tissue, even after limited liver resection [6]. The liver maximum capacity (LiMAx) test was first described by Stockmann et al. in 2009 [7]. This test is based on the intravenous application of ^13^C-methacetin, which is exclusively metabolised in hepatocytes by the cytochrome P450 1A2 enzyme into N-(4-hydroxyphenyl)-acetamide and ^13^CO_2_. The maximum liver function capacity is then calculated from the individual ^13^CO_2_/^12^CO_2_ ratio in the exhalation air of the patients.

In the past, several other liver function tests have been proposed: indocyanine green clearance (ICG), galactose elimination capacity (GEC), mono-ethyl-glycine-xylidide (MEG-X), antipyrine clearance, aminopyrine breath test (ABT) and caffeine clearance, representing quantitative liver function tests that follow different kinetics and interactions. None of them have achieved implementation in the clinical routine of patients undergoing liver resection [8].

In the clinical setting, liver function is usually measured using the values of bilirubin, albumin and prothrombin time (PT) in plasma. Liver enzymes, if elevated, provide evidence about possible liver diseases with ongoing cellular damage. However, these tests do not offer valid information about functional liver capacity due to their dependence on several factors, e.g., nutritional status, current inflammation, and cellular mass [8]. Additionally, the half-life of some of the mentioned proteins is long (several days); therefore, their informative value about the patient’s current status is limited. Liver biopsies can provide additional information about structural alterations; however, conducting liver biopsies is invasive, and there is the potential for complications as postinterventional bleeding might occur.

A current widely accepted technique to predict sufficient liver function after surgery is an estimation of the remaining organ volume—the so-called future liver remnant (FLR). The volume is assessed based on preoperative imaging by measuring the expected remaining liver volume. This volume is then typically compared to the patient’s weight. If the ratio of the FLR to body weight is ≥0.5, surgery is expected to be safe [9]. However, the volume and function of FLR do not always correlate, indicating a need for additional liver function tests [10,11,12,13]. Therefore, there is an urgent need for functionometrics in perioperative risk assessment [14,15]. Furthermore, an increasing number of patients are treated preoperatively with chemotherapy and/or PVE. The latter is a method used to induce hypertrophy in the FLR by embolising the portal vein branch of the opposite liver lobe.

The LiMAx test is, therefore, proposed to depict the current liver function as well as the potential for liver regeneration to predict the upcoming PLF [16]. However, similar to the other mentioned dynamic tests, the application of the LiMAx in the clinical routine of liver surgery has not yet been implemented broadly. In addition to other reasons, performing LiMAx tests is expensive and the cost–benefit equation is still questionable. The aim of the present study was (a) to analyse the preoperative LiMAx values of patients who underwent preoperative PVE and/or preoperative chemotherapy compared to treatment naïve patients, (b) to compare preoperative and postoperative LiMAx values in patients receiving major liver resections, and (c) to evaluate the prognostic potential of the LiMAx test as a decision-aid tool in the preoperative assessment of MLR.

## 2. Materials and Methods

The presented study was carried out as a prospective single-centre cohort study. Here, we screened and included patients who were treated at the Department of General, Visceral and Paediatric Surgery, University Medical Centre Göttingen (UMG; Göttingen, Germany) between November 2017 and September 2019. The experimental protocol was approved by the local ethics committee of the University Medical Centre Göttingen, Germany. In our clinic, the limitation of liver surgery is based on the 0.5% expected future liver remnant rule [17]. This rule was among the baselines for the in situ split publication by Schnitzbauer et al. [18]. Their data were then processed according to the suggested algorithm by Stockmann et al. [16]. Informed consent was obtained from all the included patients.

### 2.1. Study Design

We included male and female patients (≥18 years old), who were capable of giving consent, with primary or secondary liver tumours and benign lesions. The exclusion criteria were a lack of valid informed consent, people who could not speak or understand German properly, pregnant or breastfeeding female patients, patients under the age of 18 years or outpatients. A total of *n* = 40 patients were included. For all the included patients, preoperative LiMAx tests were planned. Of those, *n* = 21 patients (52.5%) were planned for MLR, and 19 patients (47.5%) were due to receive limited liver resections. For the patients with MLR, postoperative LiMAx measurements were planned on postoperative days 1, 7 and 28 for the monitoring of liver function and regeneration. Liver-specific blood parameters (PT/INR, liver enzymes, bilirubin) were scheduled for the same time points. For the patients with limited liver resections, no postoperative LiMAx tests were planned.

As the FLR was not assessed routinely in all the patients, it was calculated for this study’s purposes. However, in a few patients, the external CT scans were not adequate to calculate the FLR. According to the terms of the local ethics committee of the University Medical Centre Göttingen, Germany, an additional in-house CT scan should not have been repeated only for the purpose of calculating the FLR if the liver size was obviously sufficient for resection.

### 2.2. Application of the LiMAx Test

The application of the LiMAx test was conducted as previously described according to the recommendations by the authors and using the FLIP (“fast liver investigation package”; Humedics Ltd., Berlin, Germany) system consisting of breathing masks and ^13^C-methacetin (dose: 4 mg/mL) [16]. The exhaled individual ^13^CO_2_/^12^CO_2_ ratio was determined during a 10-min baseline measurement. The individual applied volume of ^13^C-methacetin depended on the weight of the patients (2 mg/kg body weight). The application of ^13^C-methacetin caused an increase in exhaled ^13^CO_2_, depicting the maximum liver enzymatic capacity of CYP450 1A2 for ^13^C-methacetin at this point in time. The measurement took approximately 60 min.

Every patient was instructed to abstain from beverages, food and smoking for three hours before the examination, which could bias the calculated LiMAx value. The specific oral medication of the patients was documented. We did not observe any side effects of ^13^C-methacetin in any of the patients.

### 2.3. Ethics Approval and Consent to Participate

The local ethics committee of the University Medical Centre Göttingen, Germany approved our prospective study (study number: 26/1/17). All the methods were carried out in accordance with relevant guidelines and regulations, such as the Declaration of Helsinki.

### 2.4. Statistical Anaylsis

Statistical analysis and graphs were performed using GraphPad Prism for Mac (V8.4.3). Having proofed a Gaussian normal distribution with the Anderson–Darling test, a student’s *t*-test was deployed. In case a normal distribution was not confirmed, we used a Mann–Whitney test to check for significance. The significance level was set to α = 5% for all the statistical tests.

## 3. Results

### 3.1. Patient Data

Overall, *n* = 40 patients were enrolled in the present study. The median age was 67 years (range 22–81 years). Overall, *n* = 2 out of 40 patients (5%) had benign liver lesions (haemangioma, echinococcosis), *n* = 14 patients (35.0%) were diagnosed with primary liver tumours (hepatocellular carcinoma, HCC or cholangiocellular carcinoma, CCC), *n* = 14 patients (35.0%) had liver metastases from colorectal cancer (CRC), *n* = 2 patients (5.0%) were diagnosed with Klatskin tumours, and a further *n* = 2 patients (5.0%) had gallbladder carcinoma. Overall *n* = 6 patients (15.0%) had liver metastases of neuroendocrine tumour (NET), gastrointestinal stromal tumour (GIST), malignant melanoma (MM) or adenoma of the adrenal gland (see Table 1).

For 21 patients (52.5%), MLR was performed due to an advanced oncological state. The surgical procedures included right hemihepatectomy (*n* = 9, 42.9%), extended right hemihepatectomy (*n* = 5, 23.8%), right trisectorectomy (*n* = 4, 19.0%), left hemihepatectomy (*n* = 2, 9.5%) and extended left hemihepatectomy (*n* = 1, 4.8%). For the MLR collective of patients, liver-specific blood parameters, preoperative intervention (PVE, chemotherapy) and preoperative LiMAx values are presented in Table 2.

### 3.2. Preoperative LiMAx Values

According to the algorithm of Stockmann et al. [16], medium preoperative LiMAx values (140–315 µg/kg/h) were found in *n* = 7 patients (17.5%). All the other patients (*n* = 33; 82.5%) had values greater than 315 µg/kg/h. None of the patients planned for liver resection had values less than 140 µg/kg/h. The preoperative LiMAx values did not significantly differ between the patients who underwent preoperative chemotherapy and/or PVE and those patients without any pretreatment (*p* > 0.05, see Figure 1A,B). Concerning the laboratory test, there were no significant differences in the preoperative prothrombin and bilirubin levels when comparing the pretreated patients (chemotherapy, PVE) and the therapy naïve patients (*p* > 0.05, see Figure 1C–F).

### 3.3. Pre- and Matched Postoperative LiMAx Values

The preoperative LiMAx was significantly higher than the matched postoperative LiMAx in the MLR patients (*p* < 0.0001, see Figure 2). The preoperative LiMAx and functional volumetry (FV) were used to calculate an expected value for the postoperative LiMAx testing on POD1. In *n* = 14 cases, we were able to compare an expected and an actual measured LiMAx on POD1. In these patients, the measured LiMAx values on POD 1 were both partly higher and partly lower compared to the expected values. In *n* = 7 patients (50%), the expected and measured LiMAx values differed by more than 10%. In 2 of 7 patients, the actual measured results were higher; in 5 of 7 patients, the measured results were lower than expected.

An overview of the expected and measured LiMAx values is presented in Table 3.

### 3.4. Postoperative Complications

Of 21 patients who underwent surgical resection as intended, 5 patients suffered from surgery-associated complications (patients 2, 7, 8, 15 and 17). Biliary leakage occurred in *n* = 3, impaired wound healing in *n* = 2, kidney failure in *n* = 1, multiorgan failure in *n* = 1 and severe pneumonia in *n* = 1 patients.

In detail and with respect to the LiMAx algorithm, one patient (patient 2) had a preoperative LiMAx of 531 µg/kg/h and right hemihepatectomy with relevant postoperative haemorrhage, surgical revision followed by postoperative delirium and impaired wound healing. Patient 7 (preoperative LiMAx of 325 µg/kg/h and expected LiMAx of 136 µg/kg/h after FV) had relevant postoperative biliary leakage that required endoscopic retrograde cholangiopancreaticography (ERCP) with stent implantation. Likewise, patient 8 (preoperative LiMAx 408 µg/kg/h, right trisectorectomy and expected LiMAx of 118 µg/kg/h) underwent ERCP due to postoperative biliary leakage, followed by long-term ventilation with associated pneumonia and postoperative delirium. Patient 17 (preoperative LiMAx 683 µg/kg/h and right hemihepatectomy) developed multiple organ failure and exitus letalis after stenting for biliary leakage. Histopathological analysis of this patient revealed an extensive fibrosis.

With respect to the preoperative assessment based on the LiMAx value (preoperative LiMAx 157 µg/kg/h), one [Patient 18] had an expected postoperative LiMAx of 72 µg/kg/h. This patient was a high-risk candidate, and an alternative therapy other than surgery was suggested. However, since we applied our clinical standard for decision-making and the patient did not show any laboratory alterations or morphological changes of the liver, the patient underwent extended right hemihepatectomy due to CRLM. Preoperatively, PVE was performed, and the patient had previously been treated with chemotherapy. Of interest, the measured LiMAx value on POD 1 was significantly higher (224 µg/kg/h) than the calculated value. The patient had an uneventful recovery without any signs of postoperative liver insufficiency.

### 3.5. Postoperative LiMAx Values

The patients subjected to surgery with an uneventful recovery showed a typical trend in the LiMAx values (Figure 3). There was a relevant decrease on POD 7 and a recovery on POD 28. These values generally fit into the trend of an initial decrease followed by an increase. The prothrombin time showed a comparable development. After the initial decrease, recovery can be expected.

Patient 18 with a preoperative LiMAx (157 µg/kg/h), who was encouraged to consider therapeutic options as an alternative to surgery, showed an increase at the second measurement at POD 1 (224 µg/kg/h). Unfortunately, the patient refused further analyses (therefore not shown in Figure 3).

Overall, a decrease in the LiMAx can clearly be seen in patients with prolonged recovery (see Figure 4).

## 4. Discussion

The complex physiological and biochemical processes of the liver pose a gigantic challenge in perioperative assessments of major liver surgery. Liver function mainly involves synthesis, detoxification by biotransformation, and excretion. Given this heterogeneous spectrum of functions, the assessment of perioperative liver function for predicting PLF remains a crucial issue in liver surgery. The 0.5% rule to assess resectability in liver surgery with respect to the FLR is well established [9]. In our cohort, none of the patients developed liver failure due to an undersized liver after surgery. Morbidity and consecutive mortality were all due to bile duct leaks, which are a serious challenge in liver surgery [19,20,21].

According to these results, the stratification of the LiMAx values according to the risk of liver surgery could not be evaluated. However, based on the LiMAx value, one patient was suggested for a therapeutic strategy other than surgery. The decision for surgery was based on the 0.5% rule, and this patient underwent surgery and made an uneventful recovery.

Regarding our collective of patients, a relevant portion of the patients received predominantly oxaliplatin-based preoperative chemotherapy. The LiMAx results in our study did not significantly differ between the patients with and without preoperative chemotherapy. Lock et al., however, showed significantly impaired LiMAx results in patients with preoperative chemotherapy [22]. Also, Jara et al. postulate a significant decrease in the LiMAx results in patients receiving oxaliplatin-based chemotherapy prior to liver surgery [23]. Our results are contradictory to the above-mentioned studies. One explanation might be that the regeneration of liver function is an individual process that is dependent on the individual treatment of each patient, such as the therapy regimen, cycles of chemotherapy and period of cessation of chemotherapy.

Additionally, the pretreated patients after chemotherapy did not show significant laboratory alterations (prothrombin time, bilirubin level) compared to the therapy naïve patients.

A total of 11 patients in our cohort underwent preoperative PVE 6–8 weeks before surgery. This intervention was neither associated with significant changes in the LiMAx results nor with significant laboratory changes measured one day before surgery. These results are consistent with a previous study by Malinowski et al., where a significant loss of liver function could not be observed after PVE, suggesting a buffer response from an intact hepatic artery resulting in an arterial hyperperfusion in portal hypoperfused areas [24,25]. In our study, we compared patients at 6–8 weeks after PVE to therapy naïve patients. Here, one explanation for the similar LiMAx and laboratory values could be that the hypertrophy of the FLR balanced possible PVE-induced liver function impairments.

The median of the preoperative LiMAx values was significantly higher than the median of the postoperative LiMAx values in our study, which is consistent with other previous studies hypothesising a higher liver functional capacity in more liver volume [7,16,26,27]. Nevertheless, it should be acknowledged that in some patients, there was a relevant decline postoperatively, which raised concerns, but most of these patients had an uneventful recovery. On the other hand, some patients with good postoperative LiMax values had complications after surgery and a prolonged stay.

Considering these findings together with the fact that most of the patients suffering from complications after surgery refused additional analyses, it becomes increasingly evident that LiMAx measurements depend on the patients agreeing to comply with them. To the best of our knowledge, the data retrieved from this measurement do not allow for any conclusions about the reliability of low values based on patient compliance. In contrast to high values, low values affect therapeutic strategies. Accordingly, repeating the measurements could be necessary, resulting in high costs and a decrease in patient motivation.

On the other hand, patients with an FLR below the 0.5% rule might benefit from LiMAx assessment. In these patients, high values might give confidence in a surgical approach. However, this would be a different application of the LiMAx than its intended indication in liver cancer patients, cirrhosis or fibrosis, alcohol abuse or nutritive toxic liver diseases, obesity, preoperative liver enzyme elevation or cholestasis, or after systemic or local chemotherapy as well as in patients who have received local radiation or previous liver resection. Evaluating this small subgroup of patients, however, may raise ethical concerns as there is no good rescue after PLF.

Considering the LiMAx algorithm itself, a few limitations should be addressed. First, like other dynamic liver function tests, the preoperative LiMAx test only represents the global liver function capacity so that the real functional capacity of the FLR can only be estimated approximately in the preoperative setting. In addition, as Tomassini et al. stated, this algorithm presumes homogenous functional liver parenchyma regardless of cholestasis, vascular impairment or pretreated liver tissue (i.e., PVE, chemotherapy) [28]. In our study, pretreatment of the liver tissue (PVE or chemotherapy) did not lead to significantly altered LiMAx values.

However, scintigraphic and MRI methods have shown that liver function is differentially distributed within the liver parenchyma [29,30]. Second, using this algorithm assumes that the proportion of the FLR to total liver volume correlates with the contribution of the FLR to the total liver function, as Rassam et al. indicated [14]. However, studies have shown that the FLR volume might not correlate with its function [6,31,32,33,34]. Furthermore, the LiMAx test captures one well-known pathway in the liver. 13C-Methacetin is metabolised by the enzyme cytochrome P450, particularly by family 1 subfamily A member 2 (CYP1A2). However, it has been hypothesised that CYP1A2 does not play a relevant role in the LiMAx test [35], and a recent study [36] raises relevant concerns that gene polymorphisms play a pivotal role in the inducibility of CYP1A2, resulting in unreliable LiMAx values.

Applying LiMAx measurements after surgical intervention showed limitations, as already mentioned above. As expected, the LiMAx values showed an initial decrease followed by a steady increase. These measurements were comparable to standard laboratory parameters such as prothrombin time. An exact comparison of sensitivity for recovery was not possible as the number of laboratory analyses was much higher. On the other hand, in patients with complicated postoperative development, the increase in the LiMAx values was impaired, reflecting a slower potential for regeneration. However, the development of the LiMAx values was comparable to well-established laboratory parameters. Consequently, using the LiMAx in the postoperative setting is feasible and well-tolerated. Nevertheless, it does not add any relevant information.

In conclusion, the LiMAx may have generated relevant data that reflected liver recovery, but it did not show any superior aspects over standard parameters measuring liver function. If standard criteria about liver resection with respect to the FLR are applied, the LiMAx does not add relevant information and, therefore, requires further investigation into its proper applications and justification of its costs.

## 5. Conclusions

The LiMAx values in the patients after pretreatment of the liver (via PVE and/or chemotherapy) were not significantly altered compared to those of the therapy naïve patients. The LiMAx values decreased significantly after surgery, and they did not offer any additional information on the patients who underwent extended liver surgery within the 0.5% rule. Additionally, the LiMAx may have reflected liver recovery, but it did not deliver additional information regarding postoperative liver recovery compared to conventional blood parameters. The clinical application of the LiMAx might be relevant in selected patients beyond the 0.5% rule.

### Limitations of This Study

As already indicated, patient compliance affects the LiMAx test. Additional factors, such as nutrition or smoking, have an impact on the measurements. Accordingly, noncompliance with the instructions given prior to the test may have additionally biased the measurements. To the best of our knowledge, all the patients adhered to these restrictions. All the parameters can be adjusted and optimised. However, the extent to which patients are willing to participate in the long-lasting process of LiMAx testing cannot be categorised. Moreover, there is no technical feedback loop to control patient compliance during measurements or as a reliability assessment of the LiMAx values. Finally, our cohort of patients was heterogeneous regarding diseases and pretreatment, such as PVE or chemotherapy, as well as steatotic or fibrotic tissue. These factors were not taken into account but may serve as indications for the usefulness of the LiMAx in patients with borderline FLR.

## Figures and Tables

**Figure 1 biomedicines-12-00254-f001:**
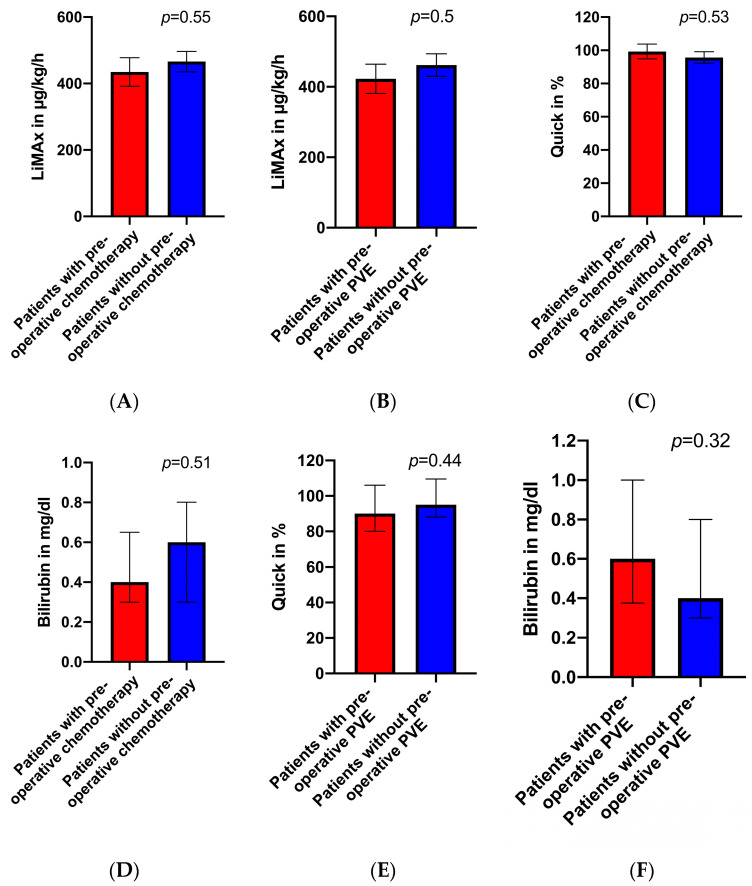
(**A**) Preoperative LiMAx values in patients with and without preoperative chemotherapy (*p* > 0.05). (**B**) Preoperative LiMAx values in patients with and without preoperative PVE (*p* > 0.05). (**C**) Preoperative Quick values in patients with and without preoperative chemotherapy (*p* > 0.05). (**D**) Preoperative bilirubin levels in patients with and without preoperative chemotherapy (*p* > 0.05). (**E**) Preoperative Quick values in patients with and without preoperative PVE (*p* > 0.05). (**F**) Preoperative bilirubin levels in patients with and without preoperative PVE (*p* > 0.05).

**Figure 2 biomedicines-12-00254-f002:**
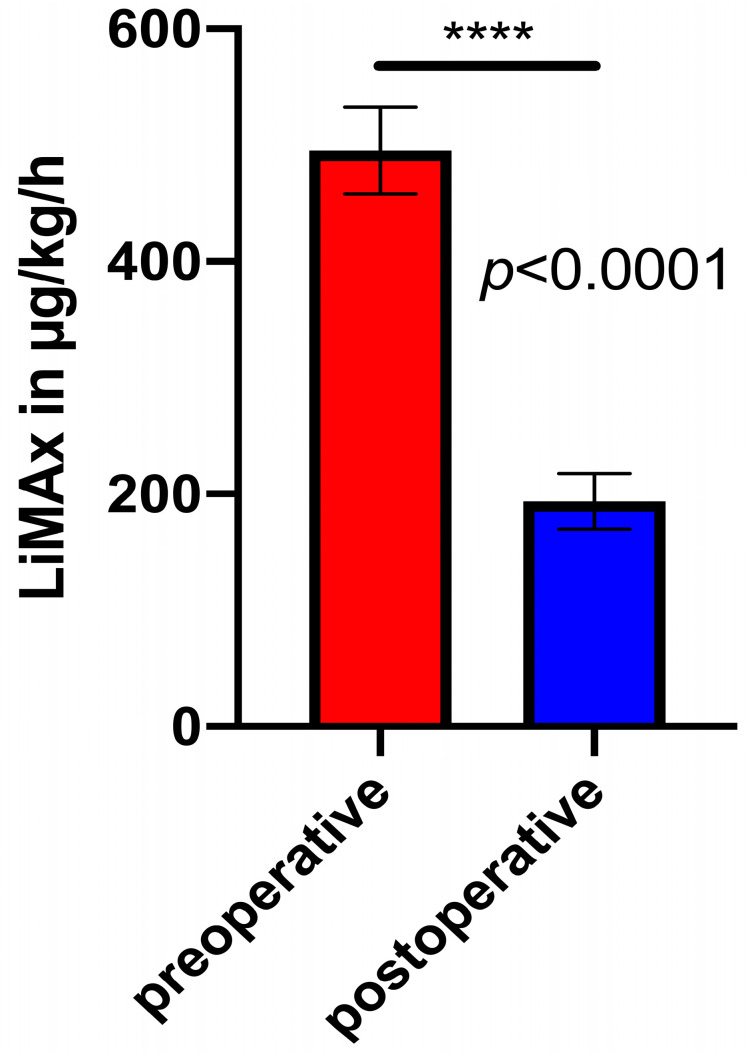
Comparison between preoperative LiMAx and matched postoperative LiMAx values on POD1 (*p* < 0.0001). ****: *p* < 0.0001.

**Figure 3 biomedicines-12-00254-f003:**
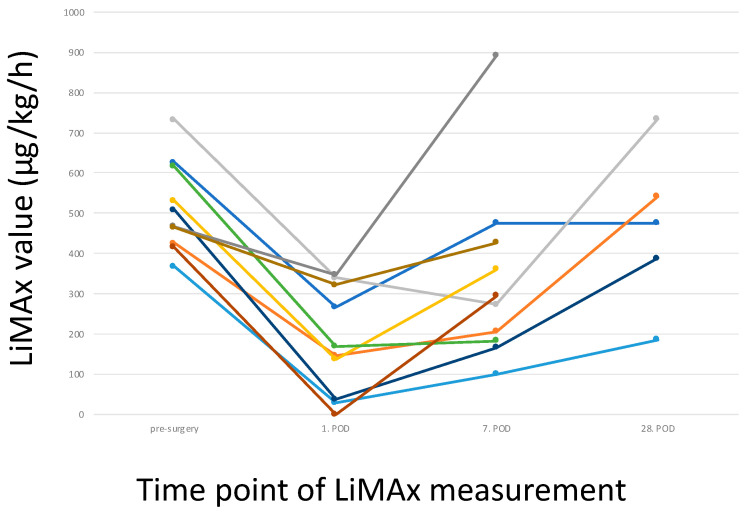
Development of LiMAx values preoperatively and postoperatively in patients undergoing liver resections without postoperative surgical complications. Each colour represents an individual patient.

**Figure 4 biomedicines-12-00254-f004:**
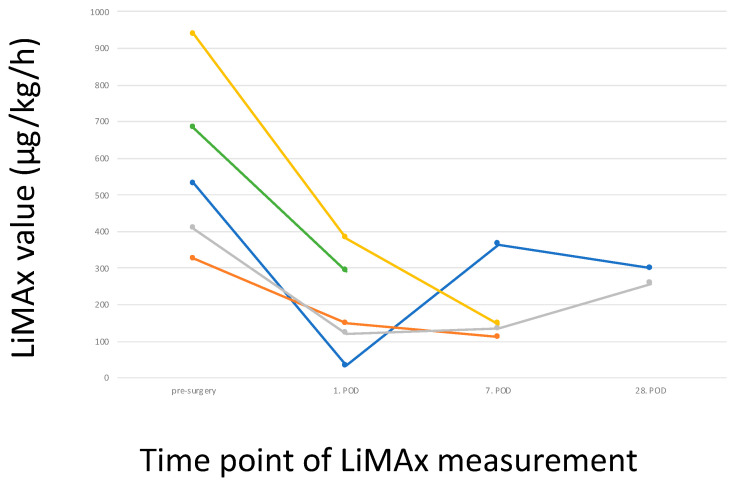
Development of LiMAx values preoperatively and postoperatively in patients undergoing liver resections with postoperative surgical complications. Each colour represents an indidivual patient.

**Table 1 biomedicines-12-00254-t001:** Overview of all included patients (*n* = 40): Benign lesions implied echinococcosis and haemangioma; CRLM: colorectal liver metastases; HCC: hepatocellular carcinoma; intrahepatic CCC: intrahepatic cholangiocellular carcinoma; GIST: gastrointestinal stromal tumour; NET: neuroendocrine tumour; MM: malignant melanoma; ICU: intensive care unit; IMC: intermediate care unit; PVE: portal vein embolisation.

Patients	*n* = 40
Age (years, median (min-max))	67 (22–81)
Sex (female/male)	21 (52.5%)/19 (47.5%)
Aetiology	
Benign, *n* (%)	2 (5.0%)
HCC, *n* (%)	6 (15.0%)
Intrahepatic CCC, *n* (%)	8 (20.0%)
CRLM, *n* (%)	14 (35.0%)
Klatskin tumour, *n* (%)	2 (5.0%)
Gallbladder carcinoma, *n* (%)	2 (5.0%)
Other (GIST, NET, MM, adenoma of adrenal gland), *n* (%)	6 (15.0%)
Stay in ICU/IMC, days, median	3
Total days in hospital, days, median	17
In-house mortality	1 (2.5%)
Preoperative PVE, *n* (%)	11 (27.5%)
Preoperative chemotherapy, *n* (%)	19 (47.5%)
Nicotine abuse	7 (17.5%)
Alcohol abuse	4 (10%)

**Table 2 biomedicines-12-00254-t002:** Clinical parameters of *n* = 21 patients who underwent MLR, including preoperative liver-specific blood parameters and LiMAx values; Age in years; preop. CTx: preoperative chemotherapy. Reference value for bilirubin: 0.3–1.2 mg/dL; reference value for prothrombin time: 80–130%. RHH: right hemihepatectomy; RTRI: right trisegmentectomy; ERHH: extended right hemihepatectomy; ELHH: extended left hemihepatectomy. n/a: not available.

Patient	Age	Volumetry Required	Disease	PVE	Preop. CTx	Surgery	Bilirubin (mg/dL)	Prothrombin Time (%)	Preop. LiMAx
1	64	no	HCC	no	no	RHH	0.3	122	627
2	55	no	HCC	no	no	RHH	0.3	95	531
3	22	yes	benign	yes	no	RTRI	0.4	90	426
4	73	yes	HCC	no	yes	RTRI	0.4	94	732
5	81	yes	NET	no	yes	ERHH	0.4	87	531
6	77	yes	CRLM	no	no	ERHH	<0.3	90	368
7	72	no	CRLM	no	yes	RHH	0.4	97	325
8	71	yes	CRLM	yes	yes	RTRI	0.6	68	408
9	37	yes	benign	no	no	RHH	0.7	82	297
10	65	no	CRLM	yes	yes	RHH	0.8	87	534
11	50	yes	CRLM	yes	yes	ELHH	n/a	100	616
12	73	yes	GIST	yes	yes	ERHH	0.3	122	509
13	48	yes	CCC	yes	no	RTRI	0.7	106	415
14	64	no	NET	no	no	LHH	0.6	98	468
15	53	no	CRLM	no	yes	RHH	0.3	150	939
16	80	no	CCC	no	no	RHH	0.8	89	612
17	71	no	CCC	no	no	RHH	0.3	85	683
18	62	yes	CRLM	yes	yes	ERHH	0.6	122	157
19	72	no	HCC	no	no	LHH	0.8	95	465
20	69	no	MM	no	no	ERHH	0.6	108	425
21	64	yes	CRLM	no	yes	RHH	0.3	113	337

**Table 3 biomedicines-12-00254-t003:** Clinical parameters concerning volumetric data. Height in cm; Weight in kg; Tumour vol.: tumour volume in cm^3^; Resected vol.: resected volume in cm^3^; FLR: future liver remnant in cm^3^; Total liver vol.: total liver volume in cm^3^; Preop. LiMAx: preoperative LiMAx; postop: postoperative; *: LiMAx on POD 3; n/a (not applicable): external CT scans were not compatible with the volumetric software (Syngo, syngo.via, 3rd, Siemens Healthineers, Erlangen, Germany) or the preoperative MRI scans; # tumour volume was estimated using a semiquantitative 5-point Likert scale.

Pat.	Total Liver Vol.	Tumour Vol. #	Resected Vol.	FLR	Preop. LiMAx	Expected LiMAx (Post-Op)	Measured LiMAx (POD 1)	Difference Expected vs. Measured LiMAx (%)
1	n/a	n/a	n/a	n/a	627	n/a	267	/
2	n/a	n/a	n/a	n/a	531	n/a	33	/
3	2444	733	1672	772	426	191	146	31
4	1971	591	1338	633	732	329	340	3
5	n/a	n/a	n/a	n/a	531	n/a	137	/
6	n/a	n/a	n/a	n/a	368	n/a	29	/
7	2970	891	2077	893	325	136	149	9
8	1514	151	1117	397	408	118	121	2
9	1022	102	665	357	297	112	141	21
10	1745	523	1084	661	534	288	192	50
11	1937	193	757	1180	616	412	169	144
12	1512	1058	1023	489	509	544	38	1332
13	1917	958	1414	503	415	215	135 *	/
14	1640	164	559	1081	468	341	347	2
15	1464	146	961	503	939	356	381	7
16	1525	152	1048	477	612	208	94	121
17	n/a	n/a	n/a	n/a	683	n/a	293	/
18	1453	145	844	609	157	72	224	68
19	1815	181	651	1164	465	330	323	2
20	n/a	n/a	n/a	n/a	425	n/a	n/a	/
21	n/a	n/a	n/a	n/a	337	n/a	n/a	/

## Data Availability

The datasets used and/or analysed during the current study are available from the corresponding author upon reasonable request.

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
