# Peer review of "Perioperative LiMAx Test Analysis: Impact of Portal Vein Embolisation, Chemotherapy and Major Liver Resection"

_biomedicines, 2024, doi:10.3390/biomedicines12020254_

Round 1
Reviewer 1 Report
Comments and Suggestions for Authors
This study provides the essential findings on the pre and postoperative liver functions using the LIMAx test to estimate the remaining organ volume. The study was performed on a heterogeneous cohort of 40 patients with different Benign lesions implied echinococcosis and haemangioma; CRLM; HCC; intrahepatic CCC; GIST; NE; MM; ICU; IMC; 1 PVE, sex, age, etc. Despite the heterogeneous cohort of a small number of patients, the findings would benefit the practitioners as a test for liver functions. Therefore, this work could be considered for publication, considering the comments below for revision:
1. The main findings of this study are based on the liver-specific blood parameters and LiMAx values of parameters of 21 patients who underwent LMR with different surgical procedures. Provide more information on whether any association between the surgical procedure and LIMAx values would be better.
2. Another question raised from this study is whether the pre and post-operation LIMAx values could be related to the patient's age. Perhaps a correlation statistic could be performed to test this hypothesis.
3. There is a concern about reliable results due to the gene polymorphism in the enzyme CYP1A2 metabolizing 13C-Methacetin. It is essential to know if LIMAx values are affected by individual genetic differences or other factors such as age, gender, status of the liver tumors, and surgical operation procedure. After this, the reliability of the LIMAx test might be significant.
4. Postoperative values decrease after 1. POD and then increase to the preoperational level in patients undergoing liver resection without postoperative complications. In contrast, LIMAX values in patients with postoperative complications after liver resection decreased. These findings need to be clarified.
5. The standard range of the preoperative LIMAx value (140-315 µg/kg/h) was said to be found in 7 patients, but there are only two patients whose values are within this range, as given in Table 2.
6. Write patient in the first column in Table 2.
7. Line 124: This sentence is incomplete.
8. Remove the title at the top of Figures 4 and 5. Moreover, provide labels for the X and Y axes.
Author Response
The main findings of this study are based on the liver-specific blood parameters and LiMAx values of parameters of 21 patients who underwent LMR with different surgical procedures. Provide more information on whether any association between the surgical procedure and LIMAx values would be better.
It is a very interesting point. We performed further analysis by comparing two surgical procedures with a distinctive difference regarding the extent of resection: the decline of LiMAx values between right hemihepatectomy (n=9) and right trisegmentectomy (n=4) was quite similar (mean decline in right hemihepatectomy to 34.5%, mean decline in right trisegmentectomy to 34.9%). In general, the LiMAx value is associated with functional liver parenchyma. In this cohort, the preoperative LiMAx value is likely to be impaired by the tumour masses. Furthermore, confounding factors like liver fibrosis/cirrhosis interfere with the LiMAx value as well. Therefore, the extent of surgical resection itself is not necessarily reflected by the postoperative LiMAx value. Besides, the present cohort is not large enough for statistical analysis between the different surgical procedures.
Another question raised from this study is whether the pre and post-operation LIMAx values could be related to the patient's age. Perhaps a correlation statistic could be performed to test this hypothesis.
Thank you for this hint. We performed a statistical analysis (please see the attachment) but we could not detect any association between age and preoperative/postoperative LiMAx values.
There is a concern about reliable results due to the gene polymorphism in the enzyme CYP1A2 metabolizing 13C-Methacetin. It is essential to know if LIMAx values are affected by individual genetic differences or other factors such as age, gender, status of the liver tumors, and surgical operation procedure. After this, the reliability of the LIMAx test might be significant.
Thank you for this important point. We totally agree that it would be absolutely relevant if LiMAx values are affected by gene polymorphisms. However, the diagnostic detection of polymorphisms prior to preoperative LiMAx measurement and imminent surgery is hardly practicable from the clinical view for now. Regarding the literature there are discrepant perspectives. While Jara et al. (Jara M, Bednarsch J, Lock JF, Malinowski M, Schulz A, Seehofer D, Stockmann M. Der LiMAx-Test: ein neuer diagnostischer Test zur Messung der aktuellen Leberfunktionskapazität [Enhancing safety in liver surgery using a new diagnostic tool for evaluation of actual liver function capacity - The LiMAx test]. Dtsch Med Wochenschr. 2014 Feb;139(8):387-91. German. doi: 10.1055/s-0033-1360061. Epub 2014 Feb 11. PMID: 24519118) state that drugs or genetic polymorphisms do not have any relevant influence on CYP1A2, others argue that members of CYP1A family are significantly downregulated in hepatocellular carcinoma (McKinnon RA, Hall PD, Quattrochi LC, Tukey RH, McManus ME. Localization of CYP1A1 and CYP1A2 messenger RNA in normal human liver and in hepatocellular carcinoma by in situ hybridization. Hepatology. 1991 Nov;14(5):848-56. doi: 10.1002/hep.1840140517. PMID: 1657755.). According to Koonrungsesomboon et al., polymorphisms could indeed play a relevant role in the inducibility of CYP1A2 which could result in unreliable LiMAx values (Koonrungsesomboon N, Khatsri R, Wongchompoo P, Teekachunhatean S. The impact of genetic polymorphisms on CYP1A2 activity in humans: a systematic review and meta-analysis. Pharmacogenomics J. 2018 Dec;18(6):760-768. doi: 10.1038/s41397-017-0011-3. Epub 2017 Dec 27. PMID: 29282363). In short, this question remains subject to further analysis and cannot be answered at the moment.
Postoperative values decrease after 1. POD and then increase to the preoperational level in patients undergoing liver resection without postoperative complications. In contrast, LIMAX values in patients with postoperative complications after liver resection decreased. These findings need to be clarified.
That is true. This finding is also subject to further analysis. However, one explanation might be that the LiMAx test reflects impairment of liver function as well as we see impairments in blood values (e.g. drop of Quick value) as sign of impaired liver function while there is an infection in a patient. The LiMAx test might therefore distinguish between patients with and without complications in the postoperative setting. Though these results need further validation in a larger cohort of patients. Nevertheless, it is a relevant observation, therefore we add these arguments in the discussion part.
The standard range of the preoperative LIMAx value (140-315 µg/kg/h) was said to be found in 7 patients, but there are only two patients whose values are within this range, as given in Table 2.
We can clarify this point. In total, 7 out of 40 patients had values between 140 and 315 µg/kg/h. We could only obtain pre- and matched postoperative LiMAx values from 21 patients. Within the 21 patients, only two patients had values between 140 an 315 µg/kg/h, as from the other five patients only preoperative LiMAx values could be obtained.
Write patient in the first column in Table 2.
"Patient" has been added in the first column in Table 2.
Line 124: This sentence is incomplete.
Thank you for your attentive reading. The sentence has been deleted in the revised manuscript.
Remove the title at the top of Figures 4 and 5. Moreover, provide labels for the X and Y axes.
We removed both titles and provided labels for the X and Y axes.

Reviewer 2 Report
Comments and Suggestions for Authors
Before doing the review, I need to state that my report is based on my experience as an engineer working in medical applications and extended collaboration with surgeons, however not from the perspective of a liver/oncology surgeon.
In the Study design, (maybe it is clear for surgeons) but as you discuss about primary and secondary tumors how do you decide whether to apply a major resection (I’m focusing here more on secondary lessions, thus a palliative treatment) instead of a less invasive one like TACE, ablation, cryoablation. These are, of course, not curative treatments but in secondary liver tumors one might consider them as a less invasive and life prolonging alternative to a major resection which does not treat the primary tumor.
Thus I would suggest to detail the reasoning of a MLR instead of a minimally invasive approach. I would expect a longer life expectancy, or even a combined curative treatment (for the liver and the primary tumors located in other body areas). Also, I do believe that it important to explain whether the location of the tumors are, in any case, an impediment to MLR or it can be performed no matter what.
The entire study, the selection of patients and the management of the entire statistical data as well as the discussions are very well documented and based on the conclusions they aimed to present the problem of MLR in its correct context and limitations.
As I worked in a three year project involving HCC patients I am aware of the complexity of liver disease and the oncologic options and the variety of treatment options each with its limitations, and based on this experience I do consider that this study is a very well conducted one and thus in my limited experience I do recommend it’s publication.
Some small observations:
In the introduction (line 75) the authors discuss the ratio between the FLR and the patient weight. Maybe a detail on the measurement units would be nice to reveal how a value over 0.5 can be obtained.
Please define PVE (Portal Vein Embolization) on its first use - abstract
Author Response
Before doing the review, I need to state that my report is based on my experience as an engineer working in medical applications and extended collaboration with surgeons, however not from the perspective of a liver/oncology surgeon.
First of all we would really like to thank you for your constructive feedback. In particular, we really appreciate that this manuscript is being reviewed by an engineer who is absolutely familiar with medical applications, devices and their complexity in the clinical daily routine.
In the Study design, (maybe it is clear for surgeons) but as you discuss about primary and secondary tumors how do you decide whether to apply a major resection (I’m focusing here more on secondary lessions, thus a palliative treatment) instead of a less invasive one like TACE, ablation, cryoablation. These are, of course, not curative treatments but in secondary liver tumors one might consider them as a less invasive and life prolonging alternative to a major resection which does not treat the primary tumor.
Thank you for this comment. From the surgical point of view, major liver resections (primary as well as secondary lesions) are only performed when there is a curative treatment option. These resections do not play any relevant role in the palliative setting. However, combining different therapy strategies (major liver resection plus ablation) are also established procedures.
Thus I would suggest to detail the reasoning of a MLR instead of a minimally invasive approach. I would expect a longer life expectancy, or even a combined curative treatment (for the liver and the primary tumors located in other body areas). Also, I do believe that it important to explain whether the location of the tumors are, in any case, an impediment to MLR or it can be performed no matter what.
This is an important point. MLR is only performed if there is a realistic chance of a curative treatment in a selected cohort of patients. As mentioned above, combination therapy strategies are appropriate concepts in order to achieve a tumor-free state for the patients. Of course, the location of the tumors are essential clinical findings as for example bilobular colorectal liver metastases usually do not justify a one-step hepatectomy but rather a multimodal therapy concept with the potential necessity of two-staged hepatectomy (chemotherapy, portal vein embolization/portal vein ligature followed by hepatectomy, ALPPS procedure etc.). Another advantage for surgery is the fact that a pathologist can tell us whether the tumour was resected in toto postoperatively, while after ablation therapy this diagnostic tool is not available.
The entire study, the selection of patients and the management of the entire statistical data as well as the discussions are very well documented and based on the conclusions they aimed to present the problem of MLR in its correct context and limitations.
Thank you for this constructive and positive feedback.
As I worked in a three year project involving HCC patients I am aware of the complexity of liver disease and the oncologic options and the variety of treatment options each with its limitations, and based on this experience I do consider that this study is a very well conducted one and thus in my limited experience I do recommend it’s publication.
Thank you very much for your recommendation.
In the introduction (line 75) the authors discuss the ratio between the FLR and the patient weight. Maybe a detail on the measurement units would be nice to reveal how a value over 0.5 can be obtained.
Of course. It is based on a study by Truant et al. from 2007 indicating that a future liver remnant to body weight ratio ≥0.5 is expected to be safe for major liver surgery without risking postoperative liver failure. For example, a patient with a weight of 80 kg needs a minimum future liver remnant of (80 kg / 2)x 10= 400 ml.
Please define PVE (Portal Vein Embolization) on its first use - abstract
Portal vein embolization has been defined on its first use in the abstract now and is further explained in the Introduction part (line 136 and 137).
Round 2
Reviewer 1 Report
Comments and Suggestions for Authors
This manuscript has been revised in line with my comments in generally, becaause most of my comments were related to sample size and statistics. Nevertheless, the response of the authors to the comments were satisfactory. All orther coments were taken into considiration in the revised version the manuscript.
All the best